# Flavonoids from Mulberry Leaves Alleviate Lipid Dysmetabolism in High Fat Diet-Fed Mice: Involvement of Gut Microbiota

**DOI:** 10.3390/microorganisms8060860

**Published:** 2020-06-07

**Authors:** Yinzhao Zhong, Bo Song, Changbing Zheng, Shiyu Zhang, Zhaoming Yan, Zhiyi Tang, Xiangfeng Kong, Yehui Duan, Fengna Li

**Affiliations:** 1Hunan Provincial Key Laboratory of Animal Nutritional Physiology and Metabolic Process; CAS Key Laboratory of Agro-ecological Processes in Subtropical Region, Institute of Subtropical Agriculture, Chinese Academy of Sciences; Hunan Provincial Engineeritng Research Center for Healthy Livestock and Poultry Production; Scientific Observing and Experimental Station of Animal Nutrition and Feed Science in South-Central, Ministry of Agriculture, Changsha 410125, China; yinzhaoz@163.com (Y.Z.); songbo52001@gmail.com (B.S.); nnkxf@isa.ac.cn (X.K.); 2Guangdong Provincial Key Laboratory of Animal Nutrition Regulation, South China Agricultural University, Guangzhou 510642, China; chamdpion@163.com; 3College of Animal Science and Technology, Hunan Agricultural University, Changsha 410128, China; 13786602169@163.com (S.Z.); yanzmmail@163.com (Z.Y.); todyfly@sina.com (Z.T.); 4Hunan Collaborative Innovation Center for Utilization of Botanical Functional Ingredients; Hunan Co-Innovation Center of Animal Production Safety, CICAPS, Changsha 410125, China

**Keywords:** FML, lipid metabolism, gut microbiota, acetic acid, microbiota transplantation

## Abstract

Here, we investigated the roles and mechanisms of flavonoids from mulberry leaves (FML) on lipid metabolism in high fat diet (HFD)-fed mice. ICR mice were fed either a control diet (Con) or HFD with or without FML (240 mg/kg/day) by oral gavage for six weeks. FML administration improved lipid accumulation, alleviated liver steatosis and the whitening of brown adipose tissue, and improved gut microbiota composition in HFD-fed mice. Microbiota transplantation from FML-treated mice alleviated HFD-induced lipid metabolic disorders. Moreover, FML administration restored the production of acetic acid in HFD-fed mice. Correlation analysis identified a significant correlation between the relative abundances of *Bacteroidetes* and the production of acetic acid, and between the production of acetic acid and the weight of selected adipose tissues. Overall, our results demonstrated that in HFD-fed mice, the lipid metabolism improvement induced by FML administration might be mediated by gut microbiota, especially *Bacteroidetes*-triggered acetic acid production.

## 1. Introduction

The health hazards of being overweight and obese have been well documented in the literature [1]. Given the increased prevalence of being overweight, obesity has become an important area of research with the goals to improve current diagnosis and treatment, with particular interest in the development of nutritional interventions and drugs to decrease the current obesity epidemic. In this context, designing more effective preventive strategies is of primary concern, since the traditional therapies for overweight and obesity (such as pharmacotherapy, surgery, and diet regulation) have several disadvantages, such as high recurrence rate, high cost, and even risk of bodily injury [2,3].

Over the years, mulberry leaves as traditional medicine have gained considerable attention and have been widely used as adjuvants for weight loss [4,5]. In this sense, the benefits of mulberry leaves reported so far include: weight loss, inhibition of adipogenesis, improvement of lipid profile, and fat depot reduction [6,7]. In addition, it has been demonstrated that the effects exerted by mulberry leaves are associated with improved brown adipose tissue (BAT) activity [6]. Mulberry leaf extract has also been demonstrated to exert antidiabetic activity in obese mice [8,9]. More interestingly, it is suggested that flavonoids from mulberry leaves (FML) may be the main functional component that exert this effect [6]. Apart from BAT activity, gut microbiota was also reported to be involved in the effects of mulberry leaves [6,7]. It has been clearly demonstrated that gut microbiota is involved in the regulation of obesity and its related metabolic disorders [10,11,12]. In particular, the community structure of gut microbiota can be reshaped by high fat diets (HFD) [13]. For instance, the ratio of *Firmicutes*/*Bacteroidetes* is increased in response to HFD feeding in both animal and human studies, which is a typical HFD-induced obesity-driven dysbiosis [14,15]. One of the most widely accepted strategies in attenuating gut dysbiosis is dietary intervention. In support, overwhelming evidence has reported that the consumption of certain functional foods and extracts, such as grape seed extract [16], blueberry [17], cranberry [18], Jamun (*Eugenia jambolana Lam*.) fruit extract [19], and capsaicin [20] could confer protective effects against obesity via regulating the community structure and diversity of gut microbiota.

Intrigued by these findings, we asked whether FML could also exert anti-obese effects by regulating gut microbiota. However, there is still a lack of direct experimental or clinical evidence to validate this hypothesis. Therefore, the current study explored the possibility that oral administration of FML may prevent the HFD-induced obesity by regulating the composition of gut microbiota. The results revealed that FML decreased body weight and lipid accumulation, reversed the whitening of BAT, and induced a dramatic shift in the gut microbiota of HFD-fed mice. Through gut microbiota transplantation and analyses of short-chain fatty acids (SCFAs), we found that improved lipid metabolism induced by FML was directly mediated by gut microbiota, and *Bacteroidetes*-mediated acetic acid production might possibly represent a protective mechanism for the beneficial events commented above.

## 2. Materials and Methods

### 2.1. Animal and Diets

All animal experiments accorded with the accepted standards of animal care and got approval from the Animal Care and Use Committee of Institute of Subtropical Agriculture, Chinese Academy of Sciences, and the ethic approval number is ISA-2017-031.

ICR female mice (6-weeks-old) were purchased from SLAC Laboratory Animal Central (Changsha, China) and housed in a controlled room maintaining a 12 h light–12 h dark cycle, with a temperature of 24 ± 2 °C and relative humidity of 45–60%. All mice had free access to food and drinking water during the experimentation and were randomly divided into three groups (*n* = 10): the control diet group (Con, containing 10% kcal fat), HFD (containing 60% kcal fat) group, and HFD+FML group (FML). The two diets (both Con and HFD) were purchased from Beijing HFK Bioscience Co. Ltd., (China). Mice in FML group received the HFD and a daily dose of FML (240 mg/kg body weight) (purity = 98.68%, Ningxia Vanilla Biotechnology Co., LTD, China) by gavage, while the other two groups received normal saline of the same volume. The dosage of FML was selected based on one previous study which demonstrated the insulin resistance-improving effects of FML (150 mg/kg body weight in rats, equal to intake of 240 mg/kg in mice) in rats with type-2 diabetic mellitus [21]. The overall protocol used in this study was presented in the Appendix A. Body weight was recorded weekly. After six weeks of FML treatment, samples including blood, adipose tissue (Inguinal white adipose tissues, iWAT; epididymal white adipose tissue, eWAT; perirenal white adipose tissue, pWAT; brown adipose tissue, BAT), liver, colonic digesta, and feces were collected [22].

### 2.2. FML Treatment for Antibiotic-Treated Mice

Mice (28.80 ± 0.30 g) were fed with the Con diet and treated with antibiotics (0.5 g/L vancomycin, 0.5 g/L ampicillin, 1 g/L gentamicin, 1 g/L streptomycin, Meilun Bio, Dalian, China) to clear gut microbiota. The antibiotics were diluted in drinking water and changed daily. After fifteen days of antibiotics treatment, the mice were randomly allocated into three groups: the Con, HFD, and HFD+FML group (*n* = 7/group). The mice were fed with the respective diets for an additional 2 weeks. Then, blood and tissue samples were collected for analyses.

### 2.3. Gut Microbiota Transplantation

Gut microbiota transplantation was performed as previously described (Appendix A) [22]. Briefly, mice originally fed with Con diets were treated with antibiotics (0.5 g/L vancomycin, 0.5 g/L ampicillin, 1 g/L gentamicin, 1 g/L streptomycin) for ten days to clear gut microbiota as the abovementioned description. Then, we used the regular water to replace the antibiotics-containing water, and transplanted the microbiota-depleted mice with donor microbiota from mice fed with Con (MTCon), HFD (MTHFD), and FML (MTFML) (treated for 6 weeks) for five days. Following transplantation, the mice received Con and HFD and regular water for another 14 days. Then, blood and tissues were collected for analyses.

### 2.4. Biochemical Analysis

Serum samples were separated after centrifugation at 3000 rpm for 10 min under 4 °C. Serum glucose, high density lipoprotein (HDL), low density lipoprotein (LDL), triacylglycerol (TG), and total cholesterol (CHOL) were measured using the Biochemical Analytical Instrument (Beckman CX4, Beckman Coulter, Germany) and commercially available kits from Roche (Shanghai, China).

### 2.5. Histological Analysis

Hematoxylin and eosin (H&E) staining of iWAT, eWAT, and BAT tissues was performed according to standard method [23]. The adipocyte size (10 fields per sample) was quantified using a DIXI3000 (Leica, Wetzlar, Germany).

### 2.6. Immunofluorescence

The protein expression of thermogenic marker uncoupling protein 1 (UCP1, Sigma, St. Louis, MO, USA, 1:100) in BAT was determined as previously described [22].

### 2.7. Oil Red O Staining

The oil red O staining of liver tissues was conducted as previously described [23].

### 2.8. RNA Extraction and Real-Time RT-PCR

RNA extraction and real-time RT-PCR were performed as previously described [24]. Briefly, total RNA was isolated from the liver, iWAT, eWAT, and BAT using TRIzol reagent (Invitrogen, USA) and treated with DNase I (Invitrogen, USA) to produce complementary DNA. Primers used in this study were designed using the Oligo 6.0 software program (Appendix A). Real-time PCR was conducted in duplicate with an ABI 7900 PCR system (ABI Biotechnology, Eldersburg, MD, USA). The relative expression of target genes was calculated by the 2^−ΔΔCt^ method [25].

### 2.9. Gut Microbiota Analysis

Feces from each group of mice were used for the gut microbiota analysis, according to the method previously described [22,23]. ACE, Chao1, Shannon, and Simpson are used to evaluate the complexity of species diversity [26]. Principal Coordinate Analysis (PCoA) was performed to obtain principal coordinates and visualization from complex, multidimensional data. Cluster analysis was conducted using a nonmetric multidimensional scaling (NMDS) plot. A linear discriminant analysis effect size (LEfSe) cladogram was adopted to identify abundant taxa that were enriched in different groups [27].

### 2.10. SCFAs

The concentrations of SCFAs (acetic acid, propionic acid, butyric acid, butyrate, isovaleric acid, and valerate) in fecal and colonic digesta samples were determined by using the Agilent 6890 gas chromatography (Agilent Technologies, Santa Clara, CA, USA) as previously described [22].

### 2.11. Statistical Analysis

Data between two groups were analyzed by unpaired *t* test (Prism 7.04; GraphPad Software, San Diego, CA, USA). Correlation analyses between microbiota and acetic acid as well as between acetic acid and weight of selected adipose tissues were also performed by Prism 7.04. All data are expressed as mean ± SEM. Probability values < 0.05 were considered as statistically significant.

## 3. Results

### 3.1. FML Reduced Body Weight and Adipose Accumulation in HFD-Fed Mice

To determine the effects of FML on HFD-fed mice, body weight and selected adipose tissues (iWAT, eWAT, and pWAT) were weighed (Figure 1A‒M). Relative to the Con-fed mice, the HFD-fed mice significantly gained more body weight (*p* < 0.001). However, compared with the HFD-fed mice, FML administration significantly decreased the body weight gain (*p* = 0.0895) (Figure 1B). As shown in Figure 1C–L, the weight of selected adipose tissues (iWAT, eWAT, and pWAT) and the ratio of these adipose tissues to body weight were markedly increased by HFD (*p* < 0.001). Similar to the trend of body weight gain, FML could reduce the weight of iWAT (*p* = 0.0607), eWAT, and pWAT (*p* < 0.01). Moreover, HFD significantly increased the mean adipocyte size of iWAT and eWAT relative to the Con (*p* < 0.001, Figure 1M). FML treatment significantly decreased the mean adipocyte size of eWAT in HFD-fed mice (*p* < 0.001), and had no significant effects on the mean adipocyte size of iWAT (*p* > 0.05).

Although liver weight remained unchanged by treatments in this study (*p* > 0.05, Figure 2A), oil red O staining results showed that mice fed with HFD displayed lipid accumulation, and FML administration attenuated this accumulation (Figure 2B). Then, we further measured serum concentrations of glucose, TG, CHOL, HDL, and LDL (Figure 2C) and found that serum levels of glucose, TG, CHOL, and HDL were significantly increased in the HFD group compared to the Con group (*p* < 0.01). FML administration decreased serum levels of TG (*p* < 0.05) and HDL (*p* = 0.0627), and exerted no effects on serum levels of glucose, CHOL, and LDL (*p* > 0.05). The quantification values of glucose, TG, CHOL, HDL and LDL were presented in Appendix A. Collectively, FML treatment (240 mg/kg body weight) for six weeks could reduce body weight gain and the lipid accumulation of HFD-fed mice.

### 3.2. FML Improves Gene Expression of Lipid Metabolism in HFD-Fed Mice

To explore the mechanism of the action of FML in lipid metabolism, we examined the expression of several genes related to lipid metabolism (ACC, PPARα/γ, SREBP1/2, and LXRα/β) in the iWAT, eWAT, and liver (Figure 3A–C). In the iWAT and eWAT, HFD significantly upregulated the mRNA expression of ACC, PPARα/γ, SREBP1/2, and LXRα/β (*p* < 0.05). However, FML treatment could reverse the upregulation of ACC, PPARγ, LXRβ (*p* < 0.05), and SREBP1 (*p* = 0.0861) triggered by HFD in the iWAT, and the upregulation of ACC, LXRα/β (*p* < 0.05), PPARα (*p* = 0.0688), PPARγ (*p* = 0.0645), SREBP2 (*p* = 0.0787) induced by HFD in the eWAT (Figure 3A,B). In the liver (Figure 3C), significant increases in the mRNA expression of ACC, PPARα, and SREBP1/2 were shown in HFD-fed mice (*p* < 0.05), whereas FML markedly downregulated the mRNA expression of ACC, PPARα, SREBP2 (*p* < 0.05), and SREBP1 (*p* = 0.0807). The mRNA expression of PPARγ and LXRα/β remained unaffected by dietary treatments.

### 3.3. FML Stimulates BAT Browning in HFD-Fed Mice

As revealed in Figure 4A, mice fed with HFD exhibited increased lipid accumulation (that is, “whitening”), and FML administration reversed HFD-induced whitening and augmented the protein expression of UCP1 (thermogenic marker) in BAT. Moreover, FML upregulated the mRNA expression of UCP1 (a thermogenic gene, *p* = 0.0653), CPT-1β (fatty acid catabolism, *p* = 0.0595), PGC-1α (mitochondrial biogenesis), and FABP5 (lipid transport) (*p* < 0.05) in HFD-fed mice (Figure 4B). These data, to some extent, suggested that FML stimulated BAT browning and thermogenesis in HFD-fed mice.

### 3.4. FML Alters the Gut Microbiota Composition in HFD-Fed Mice

The effects of gut microbiota on obesity and lipid metabolism has been well documented in the literature. Therefore, we further analyzed fecal microbiota compositions by sequencing the bacterial 16S rRNA. In the current study, high-throughput pyrosequencing of the samples generated 79,198 raw reads. 74,488 clean tags were obtained after we removed the low-quality sequences, and subsequently, these clean tags were subjected to the following analysis and clustered into OTUs. The number of OTUs was not significantly different among groups (Appendix A). As presented in Figure 5A–D, the indexes of ACE, Chao1, Shannon, and Simpson were analyzed for the α-diversity of the microbiomes. Compared to the Con, HFD augmented the indexes of ACE, Chao1, Shannon (*p* < 0.05), and Simpson (*p* = 0.0917), whereas FML tended to reverse the HFD-induced reduction in the indexes of ACE (*p* = 0.0677) and Chao1 (*p* = 0.0605). To explore overall differences in β-diversity, we further examined the intestinal microbiota structural changes using uniFrac distance-based PCoA, NMDS, and LEfSe. As shown in Figure 5E,F, a distinct clustering of microbiota composition was observed between the Con- and HFD-fed mice, and the HFD+FML group shared a similar structure to that of the other two groups. According to the LEfSe cladogram (Figure 5G), the greatest differences in taxa among all groups were shown, e.g., a greater abundance of *Firmicutes*, *Clostridia*, and *Lachnospiraceae* was found in HFD-fed mice, and a greater abundance of *Bacteroidia*, *Rickettsiales*, and *Arenimonas* was found in FML-treated mice.

The overall microbial composition at the phylum, order, and genus levels also differed among the three groups. At the phylum level (Figure 5H), a dramatic shift in the gut microbiota of mice was observed following HFD feeding, as indicated by an increased abundance of *Firmicutes* (*p* = 0.0687) and a reduced abundance of *Bacteroidetes* (*p* = 0.0559). Intriguingly, FML administration restored their levels and resulted in a significant reduction in the ratio of *Firmicutes* to *Bacteroidetes* in HFD-fed mice (*p* < 0.01). At the order level (Figure 5I), the relative abundance of *clostridiales* increased in the HFD-fed mice relative to the Con-fed mice, whereas FML administration significantly decreased its relative abundance (*p* < 0.05). At the order and genus levels (Figure 5I‒J), the relative abundance of *Bacteroidales* and *Bacteroides* were reduced in HFD-fed mice, whereas FML significantly prevented their reduction (*p* < 0.01).

### 3.5. Microbiota-Depletion by Antibiotics Fails to Prevent the Anti-Obese Effect of FML in HFD-Fed Mice

Our results mentioned above suggested that the anti-obese effects of FML might be mediated by gut microbiota. These results prompted us to perform further studies using antibiotics to clear gut microbiota (*n* = 7). As shown in Figure 6A, the body weight in the mice from HFD + antibiotics group was of similar value to mice from HFD + FML + antibiotics group, and was higher than that in the mice from Con + antibiotics group, but the difference was insignificant (*p* > 0.05). Moreover, compared to the Con + antibiotics group, the iWAT, eWAT, and pWAT weight were significantly higher in HFD + antibiotics and HFD + FML+antibiotics groups (*p* < 0.05), as is the case for serum levels of TG, CHOL, and HDL (Appendix A). However, these parameters except eWAT weight were similar between HFD+antibiotics and HFD+FML+antibiotics groups (*p* > 0.05). Taken together, these results partially indicated that FML failed to exert anti-obese effects when microbiota was depleted by antibiotics in HFD-fed mice.

### 3.6. Microbiota Transplantation from FML-Treated Mice Exerts an Anti-Obese Effect in HFD-Fed Mice

To explore whether microbiota transplantation from FML-treated mice could attenuate lipid accumulation in HFD-fed mice, we transplanted fecal samples collected from Con-, HFD-, or FML-fed mice (six weeks) into microbiota-depleted mice induced by antibiotics. Then, these microbiota-transplanted mice originally fed with Con received Con or HFD for 14 days (*n* = 7), respectively. As presented in Figure 6B, the weight of body, selected adipose tissues (iWAT, eWAT, and pWAT), and liver and serum levels of glucose, TG, CHOL, HDL, and LDL in microbiota-transplanted mice fed with Con were not affected by microbiota transplantation from mice fed with either the Con, HFD, or HMB (*p* > 0.05). Interestingly, in microbiota-transplanted mice fed with HFD (Figure 6C), the body weight of mice in MTFML group significantly decreased in comparison with the MTHFD group (*p* < 0.05). Furthermore, microbiota transplantation from FML-fed mice significantly reduced the weight of iWAT, eWAT, and pWAT (*p* < 0.05) and normalized the serum content of TG and LDL in MTHFD mice to the levels of the MTCon mice. However, no significant different was observed in the liver weight and serum levels of glucose, CHOL, and HDL among all groups (*p* > 0.05). The quantification values of Glucose, TG, CHOL, HDL and LDL were presented in Appendix A. These data indicated that the phenotypic changes observed in microbiota-transplanted mice fed with HFD were due to microbiota transplantation from FML-treated mice, which did not exert effects in microbiota-transplanted mice fed with Con.

### 3.7. Microbiota Transplantation from FML-Treated Mice Changes the Gut Microbiota Composition in HFD-Fed Mice

Then, the gut microbiota phylotypes of microbiota-transplanted and HFD-fed (MTHFD) mice were further measured by sequencing the bacterial 16S rRNA V3 + V4 region (*n* = 6). As revealed in Figure 7, alterations in the overall microbial composition at the phylum, order, and genus levels were observed in the three groups. Alterations in the relative abundances of *Firmicutes* (*p* = 0.0576) and *Bacteroidetes* (*p* = 0.0788) in MTHFD mice showed the same trends as those of HFD-fed mice. Intriguingly, microbiota transplantation from the FML group significantly normalized the elevation of the relative abundance of *Firmicutes* and the reduction in the relative abundance of *Bacteroidetes* in MTHFD mice (*p* < 0.05, Figure 7D). Meanwhile, the relative abundance of *Bacteroidales* at the order level in MTHFD mice tended to decrease relative to the MTCon mice (*p* = 0.0789), whereas the MTFML group reversed this reduction (*p* < 0.05, Figure 7E). Additionally, *Clostridiales* at the order level was significantly changed in this study (*p* < 0.05, Figure 7E). However, no significant difference was observed in the relative abundance of *Bacteroides* at the genus level and in the α-diversity (Shannon and Simpson) (*p* > 0.05, Figure 7A,B,F). Furthermore, we analyzed the gut microbiota structural changes using PCoA, and found a distinct clustering of microbiota composition between the MTCon and MTHFD mice, and a similar structure between the MTCon and MTFML mice. Overall, these data indicate that gut microbiota might be required for FML to carry out its anti-obese effects on HFD-fed mice.

### 3.8. Microbiota-Mediated Acetic Acid May Participate in the Anti-Obese Effects of FML in HFD-Fed Mice

Gut microbiota alteration is closely related to SCFAs, a major class of bacterial metabolites [28]. To further explore whether the microbiota metabolites were involved in the anti-obese effect of FML, we then analyzed fecal SCFAs in HFD-fed mice. As presented in Figure 8A, the colonic levels of acetic acid and propionic acid were significantly decreased in the HFD group relative to the control group (*p* < 0.05), whereas FML administration tended to increase the acetic acid level (*p* = 0.0908) but exerted no significant effects on the propionic acid level (*p* > 0.05) in HFD-fed mice. The quantification values for SCFA were presented in Appendix A. Moreover, the production of acetic acid was inversely associated with the relative abundance of *Firmicutes* and positively correlated with *Bacteroidetes* and Bacteroidales (Figure 8B). In addition, correlation analyses between the production of acetic acid and the weight of selected adipose tissues were performed by Pearson correlation analysis (Figure 8C). Interestingly, we found that fecal production of acetic acid was negatively associated with the weight of selected adipose tissues (iWAT, eWAT, and pWAT). These interesting data indicated that the modulation of gut microbiota composition, especially *Bacteroidetes*-mediated acetic acid, might be involved in the anti-obese effects of FML.

## 4. Discussion

Previous studies have shown that mulberry leaves could exert anti-obesity effects on HFD-induced obese mice, and these effects may be associated with their abundant functional components such as flavonoids [6]. Other studies further reported that gut microbiota might be involved in the anti-obese effects of mulberry leaves [7]. However, there is no direct evidence to support that the beneficial effects of flavonoids obtained from mulberry leaves are mediated by gut microbiota. In this current study, we investigated the effects of oral administration of FML for six weeks on the development of lipid disorder and gut microbiota. We provided direct evidence that FML blocks HFD-induced lipid metabolic disorder partially by regulating the gut microbiota.

In HFD-fed mice, oral administration of FML reduced body weight gain, fat mass, and the mean adipocyte size of eWAT. Moreover, mice fed with HFD displayed a significantly elevated level of serum TG, while oral administration of FML played a protective role and reduced the serum TG level to approximately the control level. In addition, FML administration significantly reverses mRNA expressions of lipid metabolic genes. Our current results fit well with previous studies showing that a nelumbo nucifera leaf flavonoid-rich extract is effective in alleviating body lipid accumulation and blocking obesity [29]. Similar results were obtained in streptozotocin-induced chronic diabetic rats, in which an antidiabetic effect was observed in response to long-term administration of ethanol extract (rich in flavonoids) from mulberry leaves [30]. Therefore, these results suggested a beneficial effect of FML administration on HFD-induced lipid metabolic disorder.

BAT activation can augment thermogenesis and energy expenditure, thus promoting a lean and healthy phenotype [31]. Previous studies have demonstrated that BAT activity could be enhanced by mulberry leaves, as manifested by the increased BAT/body weight ratio and upregulated mRNA expression of key genes related to BAT activation (such as UCP1, PGC-1α, and PPARγ) [6]. In good accordance with these results, we also found that FML administration reversed the HFD-induced whitening of BAT and promoted its browning, as evidenced by upregulated expression of PGC-1α at the gene level and UCP1 at both the gene and protein levels. Similarly, rutin, a kind of flavonoid with abundant content in mulberry leaves (1.10 mg/g dry powder) [6], has also been shown to enhance BAT activation in both HFD-induced obese and genetically obese mice [32]. Collectively, these results indicated that FML could enhance BAT activation, and thus, promote energy to produce heat.

An increasing body of literature has identified a relationship between alterations in the composition of gut microbiota and obesity and lipid accumulation [22,23,33]. Previous studies have reported that mulberry leaves significantly increased the relative abundances of *Bacteroidetes* and *Clostridia*, and showed a tendency to reduce the proportion of *Firmicutes* in streptozotocin-induced diabetic rats [7]. Consistent with these results, our data also showed that FML treatment induced a reduction in the relative abundance of *Firmicutes* to *Bacteroidetes* in the HFD-induced mice, with a concurrent increase in *Clostridiales*. As a consequence, flavonoids may be partially responsible for the effects exerted by mulberry leaves on gut microbiota.

To further explore whether the FML-induced microbial alteration directly relates to its anti-obese effects, we first depleted gut microbiota using antibiotics in mice. We found that antibiotics exposure failed to prevent the anti-obese effects of FML in HFD-fed mice. Since microbiota-depleted animals are less susceptible to diet-stimulated obesity and metabolic syndrome [31,34], we speculated that gut microbiota plays an important role in the effects exerted by FML on lipid metabolism and accumulation. To test our hypothesis, we then transplanted microbiota-depleted mice with microbiota from Con-, HFD-, and FML-fed mice, respectively. The results indicated that in microbiota-transplanted and HFD-fed mice, MTFML significantly decreased the weight of the body, iWAT, eWAT, and pWAT, and normalized the serum TG level in the MTHFD mice to the levels of the MTCon mice. These data suggested that similar to FML-treated mice, microbiota transplantation from FML also exerted anti-obese effects in HFD-fed mice. Moreover, in response to HFD feeding, alterations in gut microbiota composition in mice transplanted microbiota from HFD- and FML-fed mice were similar to those in HFD- and FML-treated mice. Therefore, these results suggested a causal role for microbiota in lipid metabolism and obesity. However, we have no idea how FML targets gut microbiota to exert these effects, and further investigation into the specific mechanism of the action of FML is, thus, certainly warranted.

The above interesting data prompted us to undertake further SCFAs analyses in HFD-fed mice, aiming to explore whether the anti-obese effect of FML was correlated with SCFAs. Intriguingly, we found that the reduction of fecal acetic acid in HFD-fed mice tended to be restored by FML treatment. Moreover, there is a strong correlation between acetic acid production and fat weight. Our current results fit well with previous studies showing that oral sodium acetate prevents body weight gain and lipid accumulation in HFD-fed mice [23]. Moreover, data from this study showed that there was a positive correlation between the production of acetic acid and the relative abundance of *Bacteroidetes*, which is consistent with previous studies [23]. A main way to generate acetic acid from dietary carbohydrates is driven by *Bacteroidetes* [35]. Changes in the concentration of acetic acid showed the same trends as those of *Bacteroidetes* relative abundance. Therefore, these data partially solidify the notion that FML administration affects the composition of gut microbiota (especially for *Bacteroidetes*), thus increasing the production of acetic acid to improve lipid metabolic dysfunctions induced by HFD. One proposed mechanism by which acetic acid could improve HFD-stimulated lipid metabolic dysfunctions is via GPR43 signaling, a receptor for bacterially-generated acetic acid [23,36].

In summary, our results suggest that FML treatment could attenuate adiposity of HFD-fed mice and regulate the gut microbiota, promoting a reduction of the ratio of *Firmicutes* to *Bacteroidetes* and increasing the relative abundance of *Clostridiales*. Furthermore, HFD-induced lipid disorder is augmented by microbiota transplantation from HFD-fed mice, but is alleviated by microbiota transplantation from FML-treated mice, indicating the regulatory role of gut microbiota in the protective effects of FML on HFD-induced lipid dysmetabolism. In addition, FML treatment increases fecal acetic acid level, which is negatively correlated with fat weight and positively associated with the relative *Bacteroidetes* abundance. Collectively, these data suggest that FML confers protective effects against HFD-induced lipid dysmetabolism partially via the modulation of gut microbiota and acetic acid production. It raises the possibility that FML poses great therapeutic potential in treating obesity and its complications. However, it remains to be addressed whether long-term administration of FML in obese humans and animals could exert the similar effects.

## Figures and Tables

**Figure 1 microorganisms-08-00860-f001:**
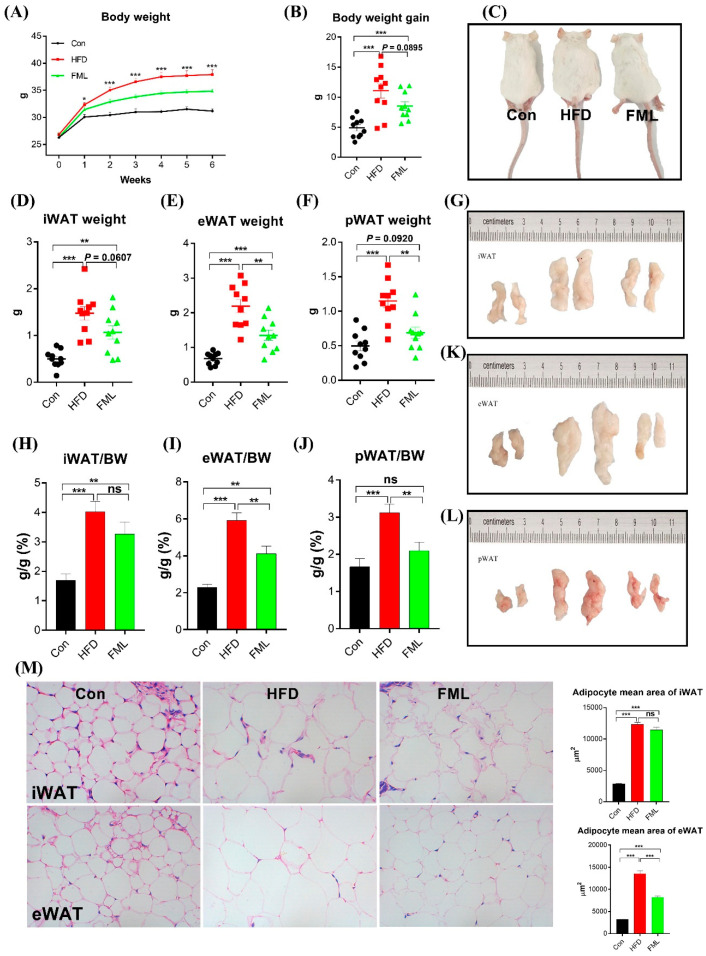
Flavonoids from mulberry leaves (FML) administration reduced body weight gain in HFD-fed mice. FML administration. (**A**) Body weight, (**B**) body weight gain, (**C**) representative mice, (**D**) inguinal white adipose tissues (iWAT weight), (**E**), epididymal white adipose tissue (eWAT weight), (**F**) perirenal white adipose tissue (pWAT weight), (**G**) representative iWAT, (**H**) iWAT/BW ratio, (**I**) eWAT/BW ratio, (**J**) pWAT/BW ratio, (**K**) representative eWAT, (**L**) representative pWAT, (**M**) H&E staining of iWAT and eWAT sections (×400) in female ICR mice fed a control diet (Con), high fat diet (HFD), or HFD with FML (FML) administration (*n* = 10) for six weeks. Data are presented as mean ± SEM, differences were denoted as follows: * *p* < 0.05, ** *p* < 0.01, *** *p* < 0.001; ns *p* > 0.05.

**Figure 2 microorganisms-08-00860-f002:**
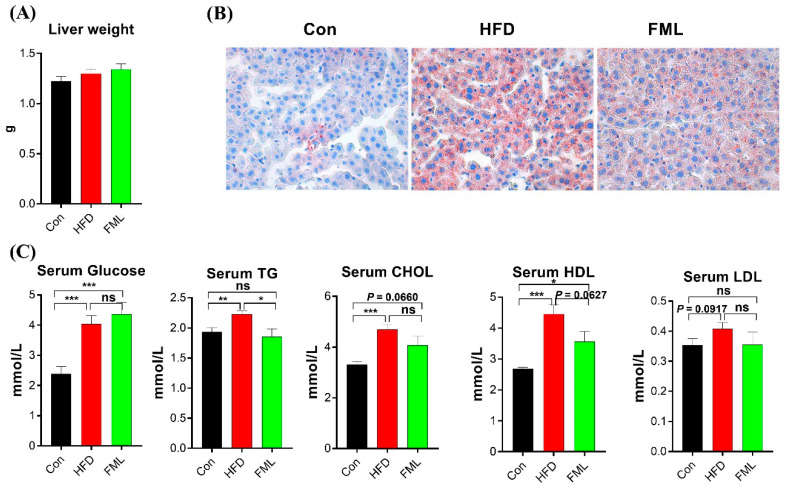
FML administration improved lipid metabolism in HFD-fed mice. (**A**) Liver weight, (**B**) liver oil red O staining (×400), (**C**) serum concentrations of glucose, triacylglycerol (TG), total cholesterol (CHOL), high density lipoprotein (HDL), and low density lipoprotein (LDL) in mice (*n* = 5). Data are presented as mean ± SEM, differences were denoted as follows: * *p* < 0.05, ** *p* < 0.01, *** *p* < 0.001; ns *p* > 0.05.

**Figure 3 microorganisms-08-00860-f003:**
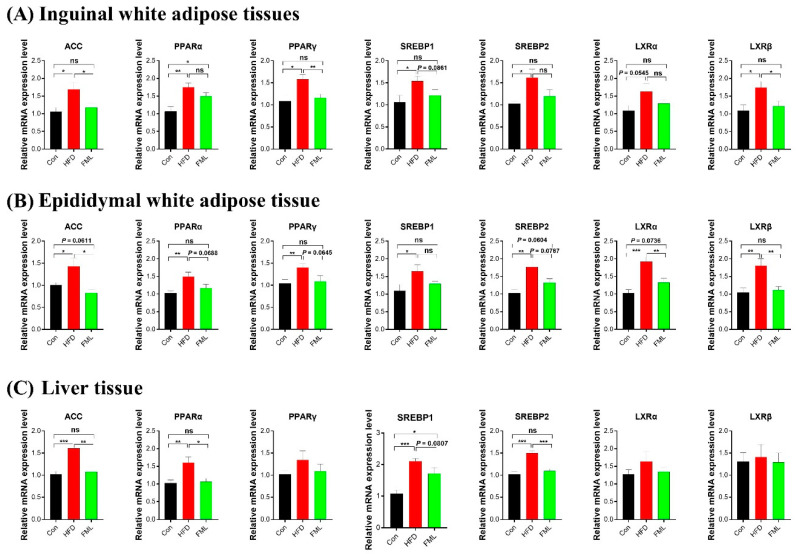
FML administration regulated the mRNA abundance of lipid metabolism-related genes in HFD-fed mice. (**A**–**C**) mRNA abundances of ACC, PPARγ, SREBP1/2, and LXRα/β in iWAT, eWAT, and liver (*n* = 8). Data are presented as mean ± SEM, differences were denoted as follows: * *p* < 0.05, ** *p* < 0.01, *** *p* < 0.001; ns *p* > 0.05.

**Figure 4 microorganisms-08-00860-f004:**
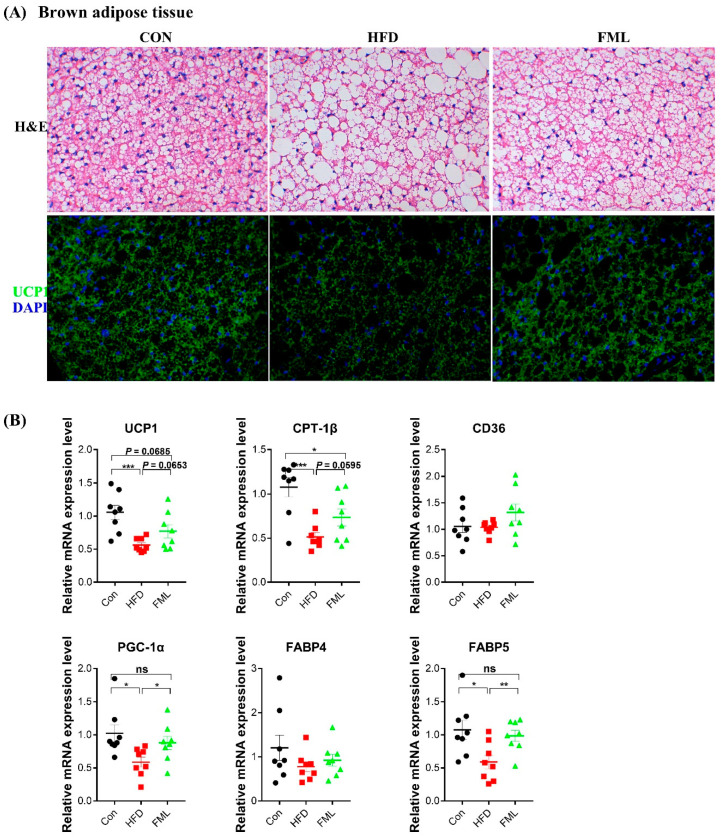
FML administration stimulated BAT browning and thermogenesis in HFD-fed mice. (**A**) H&E and UCP1 immunofluorescence staining of representative BAT sections (×400), (**B**) relative mRNA expression of markers for thermogenesis (UCP1), mitochondrial biogenesis (PGC-1α), fatty acid catabolism (CPT-1β), and lipid uptake (CD36 and FABP) in BAT (*n* = 8). Data are presented as mean ± SEM, differences were denoted as follows: * *p* < 0.05, ** *p* < 0.01, *** *p* < 0.001; ns *p* > 0.05.

**Figure 5 microorganisms-08-00860-f005:**
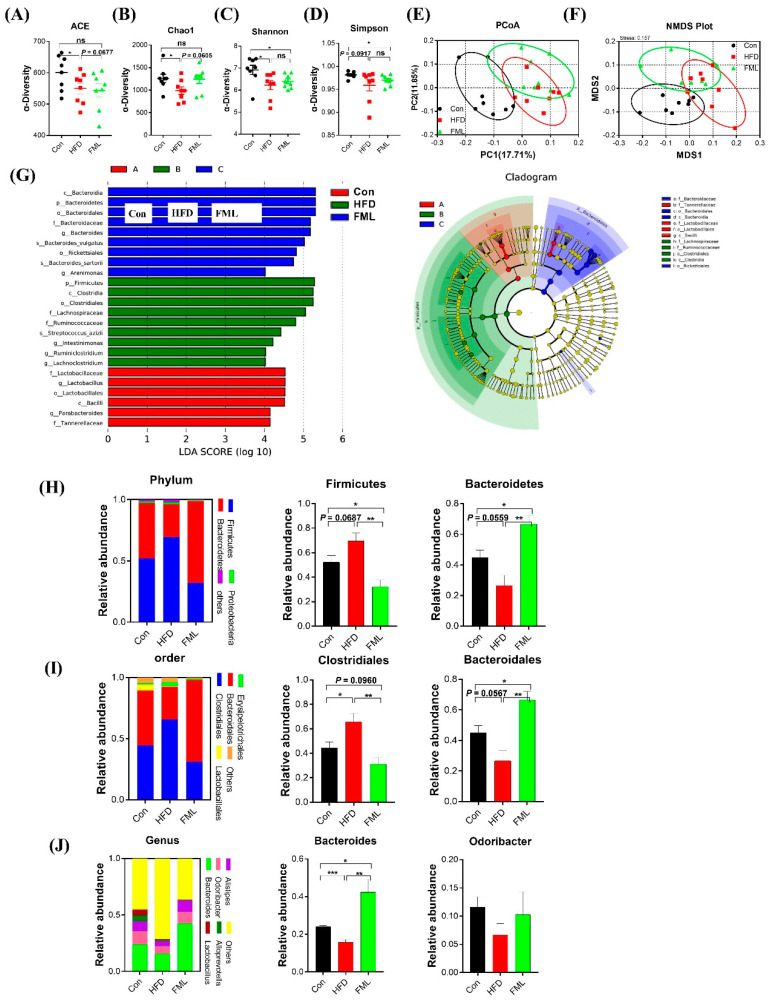
FML administration improved gut microbiota in HFD-fed mice (*n* = 8). (**A**–**D**) Indexes of ACE, Chao1, Shannon, and Simpson in α-diversity analysis, (**E**) Principal Coordinate Analysis (PCoA) plot analysis from each sample, (**F**) NMDS plot analysis from each sample, (**G**) LEfSe cladogram represents differentially abundant taxa. Only taxa with LDA scores of more than 2 are presented, (**H**) microbiota compositions at the phylum level, (**I**) microbiota compositions at the order level, (**J**) microbiota compositions at the genus level. Data are presented as mean ± SEM, differences were denoted as follows: * *p* < 0.05, ** *p* < 0.01, *** *p* < 0.001; ns *p* > 0.05.

**Figure 6 microorganisms-08-00860-f006:**
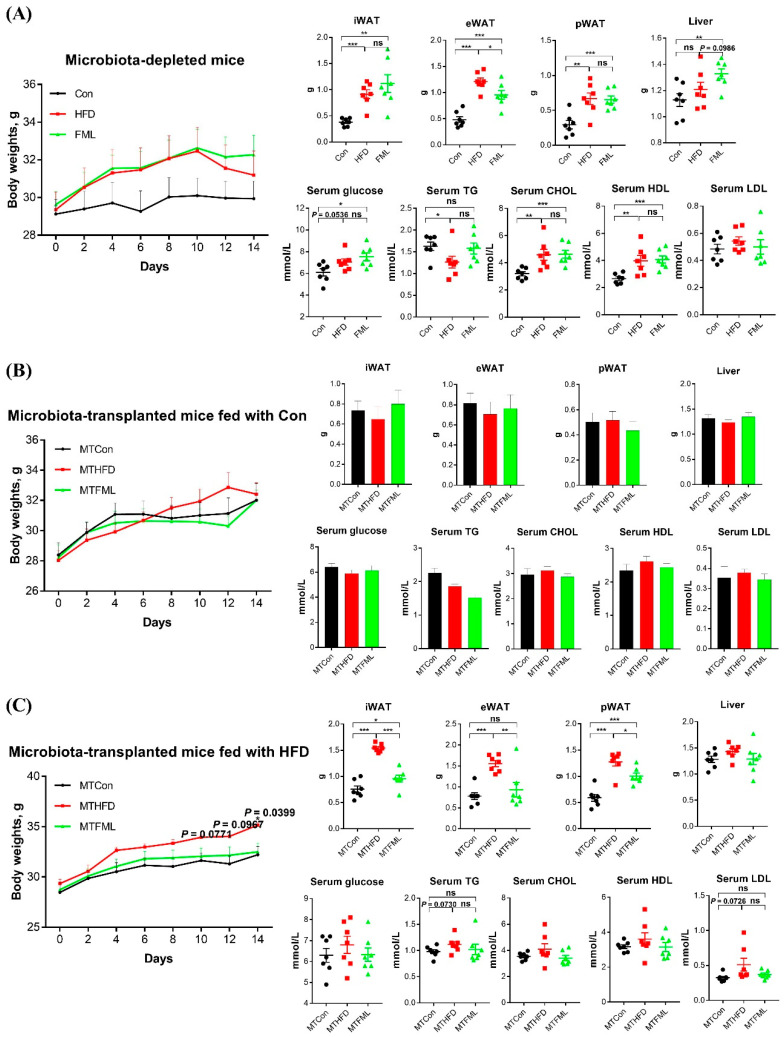
Microbiota deficiency failed to prevent the anti-obese effect of FML in HFD-fed mice, but microbiota transplantation from FML-treated mice improved lipid metabolism in HFD-fed mice but not in Con-fed mice (*n* = 7). Body weight, liver weight, iWAT weight, eWAT weight, pWAT weight, and serum levels of glucose, TG, CHOL, HDL, and LDL in microbiota-depleted mice (**A**), microbiota-transplanted mice fed with Con (**B**), and microbiota-transplanted mice fed with HFD (**C**). Data are presented as mean ± SEM, differences were denoted as follows: * *p* < 0.05, ** *p* < 0.01, *** *p* < 0.001; ns *p* > 0.05.

**Figure 7 microorganisms-08-00860-f007:**
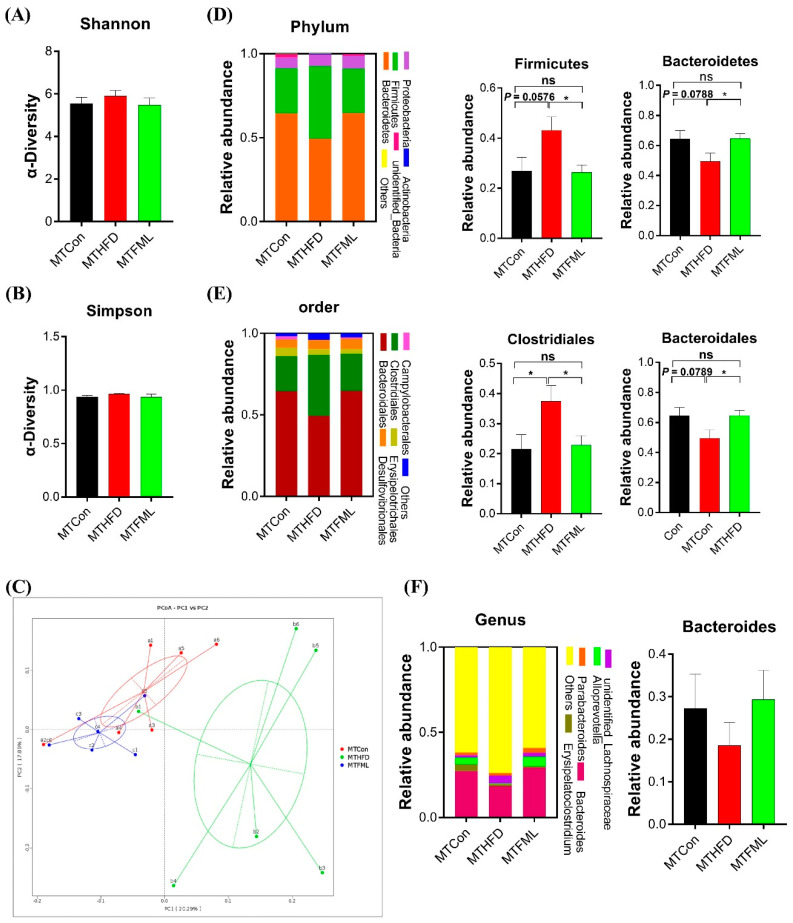
Gut microbiota in response to microbiota transplantation from Con (MTCon), HFD (MTHFD), and FML (MTFML) groups (*n* = 6). (**A**,**B**) Indexes of Shannon and Simpson in α-diversity analysis, (**C**) PCoA plot analysis from each sample, (**D**) microbiota compositions at the phylum level, (**E**) microbiota compositions at the order level, and (**F**) microbiota compositions at the genus level. Data are presented as mean ± SEM, differences were denoted as follows: * *p* < 0.05, ns *p* > 0.05.

**Figure 8 microorganisms-08-00860-f008:**
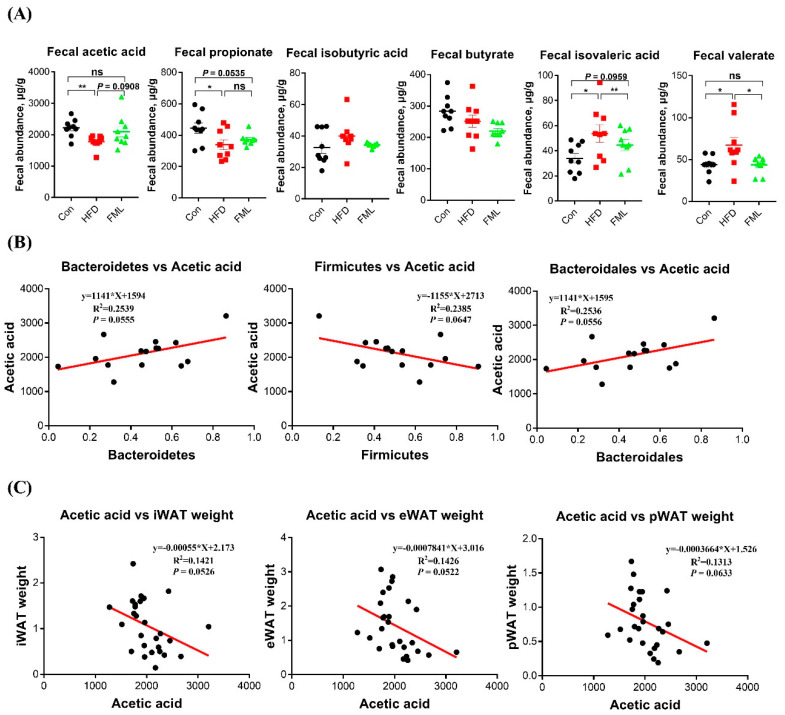
Fecal SCFAs concentrations and correlation analyses between acetic acid and altered microbiota. (**A**) Fecal acetic acid, propionate, butyrate, isobutyric acid, isovaleric acid, and valerate. (**B**) Correlation analyses between acetic acid and *Bacteroidetes*, *Bacteroidales*, and *Firmicutes*, respectively. Fecal SCFAs were measured by gas chromatography. (**C**) Correlation analyses between acetic acid and the weight of iWAT, eWAT, and pWAT, respectively. Data are presented as mean ± SEM, differences were denoted as follows: * *p* < 0.05, ** *p* < 0.01, ns *p* > 0.05. Correlation analysis between microbiota and acetic acid was conducted by GraphPad Prism 7.04.

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
