# Peer review of "Flavonoids from Mulberry Leaves Alleviate Lipid Dysmetabolism in High Fat Diet-Fed Mice: Involvement of Gut Microbiota"

_microorganisms, 2020, doi:10.3390/microorganisms8060860_

Round 1

Reviewer 1 Report

The manuscript by Zhong et al. entitled "Flavonoids from mulberry leaves alleviates lipid dysmetabolism in high fat diet-fed mice: Involvement of gut microbiota" describes a nutrient study in a mouse model. The authors have analyzed the role of the microbiome in mice fed with flavonoids from mulberry leaves (FML) on lipid metabolism in high-fat diet mice.

The study includes analysis of body weight, tissue analysis, microbiota phyla composition, lipid concentration, SCFA concentration for three different groups: Control, High fat diet and high fat diet but supplemented with FML. The author identify differences and a beneficial effect of FML treated mice with most investigated data points that the FML treatment leads to a clear and significant differences in FML-treated mice (similar to the control values) compared to the high fat diet (HFD) mice. The authors than analysed antibiotic treated mice to identify the impact of microbiota on FML treated HFD mice.

I summary, the study has been performed very well and the manuscript written well. Especially, the analysis of various models tested by the author completes this study and I only have a few remarks. My main criticisms are the lack of description of the presented figures, low quality of figures that I cannot read and which limits my assessment possibility on the statistics and missing experimental parts and raw data. I recommend publication in Microorganisms of this suitable study after major revisions.

Major remarks:

  • Figures 2, 3 and 4 cannot be read and are too low in resolution and size. I need to see optimised figures to assess the data of these important data sets
  • Page 6; Lines 3-5: The authors describe to determine lipid concentrations. Two issues about this: a) glucose is no lipid and b) experimental data on the quantification is missing.
  • The quantification values for SCFA and lipid + glucose concentrations should be provided in the SI
  • The paragraph describing Figure 1 is not descriptive and must be extended
  • The ethical approval number must be included

Minor remarks:

  • Figure 1 has two "K" annotations

Author Response

Major remarks:

  1. Figures 2, 3 and 4 cannot be read and are too low in resolution and size. I need to see optimised figures to assess the data of these important data sets

A: We have improved these figures according to the reviewer’s suggestion.

  1. Page 6; Lines 3-5: The authors describe to determine lipid concentrations. Two issues about this: a) glucose is no lipid and b) experimental data on the quantification is missing.

A: Thanks for the reviewer’s suggestion. (1) The sentence has been rewritten according to the suggestion, which is also shown as follows: “Then, we further measured serum concentrations of glucose, TG, CHOL, HDL, and LDL (Fig. 2C) and found that serum levels of glucose, TG, CHOL, and HDL were significantly increased in the HFD group compared to the Con group (P < 0.01)”.

(2) As described in the manuscript, serum concentrations of glucose, TG, CHOL, HDL, and LDL were determined using the Biochemical Analytical Instrument (Beckman CX4). Test data can be used directly without calculation, which has been widely reported (Yin J, Li YY, Han H, et al. Melatonin reprogramming of gut microbiota improves lipid dysmetabolism in high fat diet-fed mice. J Pineal Res, 2018, 65: e12524. Duan YH, Zhong YZ, Xiao H, et al. Gut microbiota mediates the protective effects of dietary beta-hydroxy-beta-methylbutyrate (HMB) against obesity induced by high-fat diets. FASEB J, 2019, 33: 10019-10033).

  1. The quantification values for SCFA and lipid + glucose concentrations should be provided in the SI.

A: Thanks for the reviewer’s suggestion. (1) Both histogram and scatter chart are the representation of data. Moreover, the scatter diagram can show more clearly the dispersion of data within the group. Therefore, we chose the scatter diagram. (2) We have provided the quantification values for SCFA and lipid + glucose concentrations in Supplementary Table 2.

  1. The paragraph describing Figure 1 is not descriptive and must be extended.

A: Thanks for the reviewer’s suggestion. We have improved this paragraph.

  1. The ethical approval number must be included.

A: The ethic approval number is ISA-2017-031, which has been added in the manuscript.

Minor remarks:

  1. Figure 1 has two "K" annotation

A: Thanks for the reviewer’s suggestion. The second “K” has been changed to “M”.

Reviewer 2 Report

In this study, the authors investigated the roles and mechanisms of flavonoids from mulberry leaves on lipid metabolism in HFD-fed mice. As an interesting and promising approach, the results demonstrated that in HFD-fed mice, the lipid metabolism improvement induced by FML administration might be mediated by gut microbiota, especially bacteroidetes-triggered acetic acid production. The article is globally well written and the results are interesting and bring new scientific data in the field of microbiota and metabolic disturbances. 

This reviewer has a few remarks about the presentation that really needs to be improved:

Introduction:

line 15: change "dysbosis"

lines 26-30: it is not common to present part of the results at the end of the introduction

line 28: change "bacterodetes"

Methods: 

This reviewer would have appreciated to see a fourth group in the protocol design: animals treated only with the FML to check the impacts on lipid metabolism without HFD. Please justify this serious flaw. 

page 3: regarding the number of animals it could be easier to specify the n/group

It could be useful to present a figure describing the overall protocol used in this study. Even the description of the methods refer to previously published materials, it would be useful to present a few details. Please modify this part.

Results: 

-In the graphs representation, please respect the same color code for each group, as for figure A and the others (black, red, green vs blue)? According to this, please change figure A. 

-The quality of photos from WAT is very poor? The authors need to improve drastically the overall quality of photos from WAT and liver.

-Most of the figures are illegible and need to be enlarged and of a much better graphic resolution if only for the analysis of the data. This reviewer is incapable of dissecting the results reliably and accurately. This reviewer would really appreciate to make a new review once the figures and graphs are improved.

Figure 4: change "microboita" in the title

Figure 6B: Bacteroidetes and bacteroidales figures appear completely similar? please justify.

The discussion is well conducted and describes the results well in relation to the bibliography of the field and the interpretation is quite pleasant.

Author Response

  1. line 15: change "dysbosis"

A: Thanks for reviewer’s suggestion. The “dysbosis” has been changed to “dysbiosis”.

  1. lines 26-30: it is not common to present part of the results at the end of the introduction

A: Thanks for reviewer’s suggestion. We agree with the idea that it is not common to present part of the results at the end of the introduction. However, over the years, in order to make readers understand the conclusion of this study more intuitively, more and more manuscirpts adopt this writing method.

  1. line 28: change "bacterodetes"

A: Thanks for reviewer’s suggestion. The “bacterodetes” has been changed to “bacteroidetes”.

  1. This reviewer would have appreciated to see a fourth group in the protocol design: animals treated only with the FML to check the impacts on lipid metabolism without HFD. Please justify this serious flaw.

A: Thanks for reviewer’s suggestion. Our previous studies have showed that FML could reduce average backfat thickness and the fat percentage of finishing pigs, suggesting that under normal conditions, FML could also improve lipid metabolism. Therefore, we did not set a control group to supplement FML in this experiment.

  1. page 3: regarding the number of animals it could be easier to specify the n/group

A: Thanks for reviewer’s suggestion. This sentence has been rewritten according to the suggestion, which is also shown as follows.

“After fifteen days of antibiotics treatment, the mice were randomly allocated into three groups: the Con, HFD, and HFD+FML group (n=7/group)”.

  1. It could be useful to present a figure describing the overall protocol used in this study. Even the description of the methods refer to previously published materials, it would be useful to present a few details. Please modify this part.

A: Thanks for reviewer’s suggestion. We have modified it according to your suggestion. The overall protocol used in this study was presented in the Supplementary Fig. 1. The dosage of FML was selected based on one previous study which demonstrated the insulin resistance-improving effects of FML (150 mg/kg body weight in rats, equal to intake of 240 mg/kg in mice) in rats with type-2 diabetic mellitus (Mu, X.Y.; Li, X.J. Influence of mori folium flavonoid on β-cells in rats with type-2 diabetic mellitus. Chinese Journal of Experimental Traditional Medical Formulae 2013, 19, 213-216.). Then, we investigated the underlying mechanisms of the action of FML, and we found that FML treatment could improve the gut microbiota composition and increase acetic acid content in HFD-fed mice. To explore whether the anti-obese effect of FML in HFD-fed mice was mediated, at least in part, by gut microbiota, gut microbiota transplantation was conducted.

  1. In the graphs representation, please respect the same color code for each group, as for figure A and the others (black, red, green vs blue)? According to this, please change figure A.

A: Thanks for reviewer’s suggestion. We have modified it according to your suggestion.

  1. The quality of photos from WAT is very poor? The authors need to improve drastically the overall quality of photos from WAT and liver.

A: Thanks for reviewer’s suggestion. We have modified it according to your suggestion.

  1. Most of the figures are illegible and need to be enlarged and of a much better graphic resolution if only for the analysis of the data. This reviewer is incapable of dissecting the results reliably and accurately. This reviewer would really appreciate to make a new review once the figures and graphs are improved.

A: Thanks for reviewer’s suggestion. We have modified it according to your suggestion.

  1. Figure 4: change "microboita" in the title

A: Thanks for reviewer’s suggestion. The “microboita” has been changed to “microbiota”.

  1. Figure 6B: Bacteroidetes and bacteroidales figures appear completely similar? please justify.

A: Thanks for reviewer’s suggestion. Bacteroidetes and bacteroidales belong to two different levels and bacteroidetes include bacteroidales. We have drawn these two graphs based on their respective real data.

  1. The discussion is well conducted and describes the results well in relation to the bibliography of the field and the interpretation is quite pleasant.

A: Many thanks for your encouragement.

Round 2

Reviewer 2 Report

Thanks for taking into account my recommendations and comments.